# Longitudinal Sexting Research: A Systematic Review

Yunhao Hu [ID], Elizabeth Mary Clancy [ID] and Bianca Klettke *

School of Psychology, Deakin University, Geelong, VIC 3200, Australia; houson.hu@deakin.edu.au (Y.H.); elizabeth.clancy@deakin.edu.au (E.M.C.)

* Correspondence: bianca.klettke@deakin.edu.au

**Abstract:** The exchange of intimate messages, images, and videos via digital means, also referred to as sexting, has drawn considerable academic attention in recent years. Specifically, cross-sectional research has indicated that sexting can be associated with harmful outcomes such as depression, anxiety, and cyberbullying. However, there is currently limited empirical research examining the causal relationship between these factors, and to date, there has been no systematic review of the longitudinal studies on sexting. Thus, the purpose of this review is to summarise and review the current research addressing long-term outcomes and predictors of sexting. A systematic search of databases was conducted. Eight databases were searched, with twenty-four longitudinal studies meeting the inclusion criteria and thus included in this review. The quality of individual studies was assessed using the Joanna Briggs Institute Critical Appraisal tool. Overall, longitudinal research into sexting is scarce, and variability in definitions, measured variables, and sample demographics have created challenges in achieving consensus across variables. For example, findings were inconclusive regarding causal relationships between sexting, cyberbullying/bullying, and psychological health outcomes. Findings indicated that positive peer norms predicted sexting and that sexting was predictive of future offline sexual behaviours. Future longitudinal research would benefit from differentiating between consensual and non-consensual sexting behaviours in measurement. Future prevention efforts should focus on addressing peer norms that develop around sexting behaviours.

**Keywords:** sexting; systematic review; harmful online behaviours; longitudinal

## 1. Introduction

With progressive advancement in digital technology in recent years, online sexual interactivity has similarly evolved [1], and sexting has accrued mounting attention in the public consciousness and the scientific literature. Although sexting is commonly characterized as the sending, receiving, or sharing of sexually explicit messages, images, or videos through electronic means [2], there is currently a lack of consensus in definitions in scientific research [3,4]. This includes incongruity with respect to included behaviours (sending, receiving, forwarding) and the type of media formats (image, text, video).

A systematic review conducted by Klettke et al. [2] estimated the mean prevalence of sext-sending in adolescents to be 11.96%, while a separate review [5] estimated slightly higher rates at 14.8%, and a review by Barrense-Dias et al. [3] found that sext-sending rates ranged from 2.5% to 27.6%. Concerning sext-receiving behaviours in adolescents, systematic reviews found that mean prevalence rates were estimated to be 15.6% [2] and 27.4% [5], with a range of 7.1% to 60% [3]. Finally, the average rate of non-consensual sext-forwarding in adolescents was found to be 12% [5].

Sexting rates tend to increase with age, with adolescents reporting lower rates of engaging in sexting behaviours than adults [2]. A systematic review by Klettke et al. [2] found an estimated adult prevalence of 58.0% for sext-receiving and 48.6% for sext-sending. More recently, a meta-analysis [6] found slightly lower rates of 41.5% for sext-receiving, 38.3% for sending, and 15% for non-consensual forwarding. Overall, text-based sexting has been found to be more common compared to image/video-based sexting [2,3]. Prevalence

rates are also higher in more recent studies, likely due to the increasing availability of personal mobile devices with cameras [5,6].

The vast majority of sexting research has focused on the outcomes and consequences of sexting. Research has been based predominantly on cross-sectional research and findings have been inconsistent, with substantial debate in the scientific community as to whether sexting should be conceptualized as deviant or normative behaviour. There is an emphasis in the literature on the harm to adolescent populations, especially adolescent girls [7]. On one hand, research suggests a myriad of negative consequences associated with sexting behaviour, including risky sexual behaviours [6], negative psychosocial consequences including depression, anxiety, and unwanted sexual solicitation [4,8–10], cyberbullying victimization [11–13], and in-person bullying, otherwise known simply as bullying [14]. On the other hand, consensual sexting in both adolescents and young adults can be healthy [15,16], utilized as a form of sexual experimentation, and bypass risks associated with having sex, such as unplanned pregnancies and STDs [17]. Some researchers argue that conceptualizing sexting as a risky behaviour can perpetuate victim blaming, given that responsibility is thereby placed on victims of online abuse to avoid consensual sexting, a common method of sexual expression among youth [7].

Further, risks traditionally associated with sexting behaviours, such as cyberbullying and internalizing problems, are more closely associated with the non-consensual forms of sexting behaviours, such as the non-consensual dissemination of intimate images [18–22]. Pertinently, victims of non-consensual sext-sharing are not only prone to experience subsequent blackmail, bullying, and internalizing problems previously associated with sexting generally [18,20,23,24], but also more vulnerable to intimate partner violence [25], mental trauma, and suicidal ideation [26–28].

Whilst there is a broad scope of cross-sectional literature in sexting research, it is difficult to establish causal relationships, given few longitudinal studies and a limited focus on sexting behaviours and their relationship with associated factors such as psychosocial adjustment. Thus, a better understanding of extant research is warranted to guide future research. To date, several reviews have summarized findings from prior studies on sexting behaviours [2,5–7,26]; however, these have been largely focused on prevalence rates and motivations. To the authors' knowledge, there are currently no reviews that have systematically reviewed longitudinal studies investigating sexting behaviours. Therefore, the aim of this paper was to collate and summarise existing evidence in longitudinal research on sexting behaviours and associated factors through a systematic review.

## 2. Materials and Methods

### 2.1. Study Selection Criteria

Articles were considered for this review if the following criteria were satisfied: the article utilized a longitudinal design; the article investigated sexting behaviours, where sexting was defined as the sending, receiving, or forwarding of intimate texts, images, or videos via digital means; the study reported relationships with other behavioural or attitudinal variables (e.g., mental health, sexual activity, substance use, cyberbullying, norms, and attitudes); the article presented original findings, rather than a summation or critique of previously reported data; the article presented sufficient data so that the methods and results may be extracted and analysed; and the article was peer-reviewed and published in English. Given the paucity of existing research in longitudinal sexting behaviour, to maximise the body of research within the current review, no exclusion criteria were imposed on population or study length. The review strategy followed PRISMA guidelines [29], and a completed PRISMA checklist is included as Figure S1.

### 2.2. Search Strategy

On 21 November 2022, the following electronic databases were searched in the title, abstract, and keyword fields: Academic Search Complete, CINAHL, MEDLINE, EMBASE, PsycINFO, PsycArticles, PsycExtra, and Psychology and Behavioural Sciences Collection. Search terms were identified by referencing prior systematic reviews of sexting research [2,4–6] and using synonyms for the following terms: "sexting", "non-consensual sext dissemination", and "longitudinal". For example, non-consensual sext dissemination has been commonly referred to in research as image-based sexual abuse [30] or technology-facilitated sexual violence [31]. These keywords were chosen to encapsulate the broad scope of definitions and conceptualizations in current sexting research. A summary of the search syntax is presented in Table 1. Finally, the reference lists of other sexting reviews known to the authors were searched for relevant articles.

**Table 1.** Search syntax for systematic review.

| Outcome | Descriptor | |
|---|---|---|
| Search for Sexting | 1. | Sext |
| | 2. | Sexting |
| | 3. | Sexts |
| | 4. | Nudes |
| | 5. | Selfie |
| | 6. | intimate W3 (image OR photo OR picture OR messag *) |
| | 7. | sex * W3 (image OR photo OR picture OR messag *) |
| | 8. | explicit W3 (image OR photo OR picture OR messag *) |
| | 9. | private W3 (image OR photo OR picture OR messag *) |
| | 10. | OR/1–9 |
| Search for Non-Consensual Sexting | 11. | revenge porn * |
| | 12. | non-consensual porn * |
| | 13. | involuntary porn * |
| | 14. | online sexual |
| | 15. | image W3 abuse |
| | 16. | image W3 violence |
| | 17. | image W3 sexual |
| | 18. | technology W3 abuse |
| | 19. | technology W3 violence |
| | 20. | OR/11–19 |
| Search for Longitudinal | 21. | Longitudinal |
| | 22. | repeated measure |
| | 23. | follow up |
| | 24. | OR/21–23 |
| Final Search | 25. | (10 OR 20) AND 24 |

Note: * = wildcard symbol used to broaden the search by finding terms that start with the same letters (e.g., message, messages, messaging); W3 = wildcard notation used to find terms within 3 words of each other (e.g., find image within 3 words of abuse).

The initial search returned 1318 articles, with 594 remaining after duplicate removal. These articles were screened by title and abstract by the first author. As noted by Waffenschmidt et al. [32], while second screening is typically recommended, it may result in minimal gains in comparison to the resources required for full double screening, particularly with an experienced reviewer. Given resource constraints, a full double screening of titles and abstracts was not possible. However, a random sample of 10% was reviewed by the second author for verification purposes, with no discrepancies noted in eligibility, providing confidence in the screening process.

Screening resulted in 31 articles for full-text analysis, of which 22 studies were retained. Studies were rejected if they were not directly related to sexting behaviours (*n* = 6), had insufficient data, had no original findings, or had no longitudinal study design (*n* = 1 each). Two additional studies were identified by manually examining reference lists of existing reviews, leaving a total of 24 studies for inclusion in the final review. A summary of the identification, screening, and inclusion process is presented as a PRISMA flow diagram

in Figure 1. Data was then extracted by the first author and a risk of bias assessment was conducted using the JBI Critical Appraisal checklist by the first and second authors (Table S1). Given the significant variation in the operationalization of included variables and heterogeneity in research design, a meta-analysis was considered but deemed impossible. Instead, conceptual findings were extracted and presented in the current review.

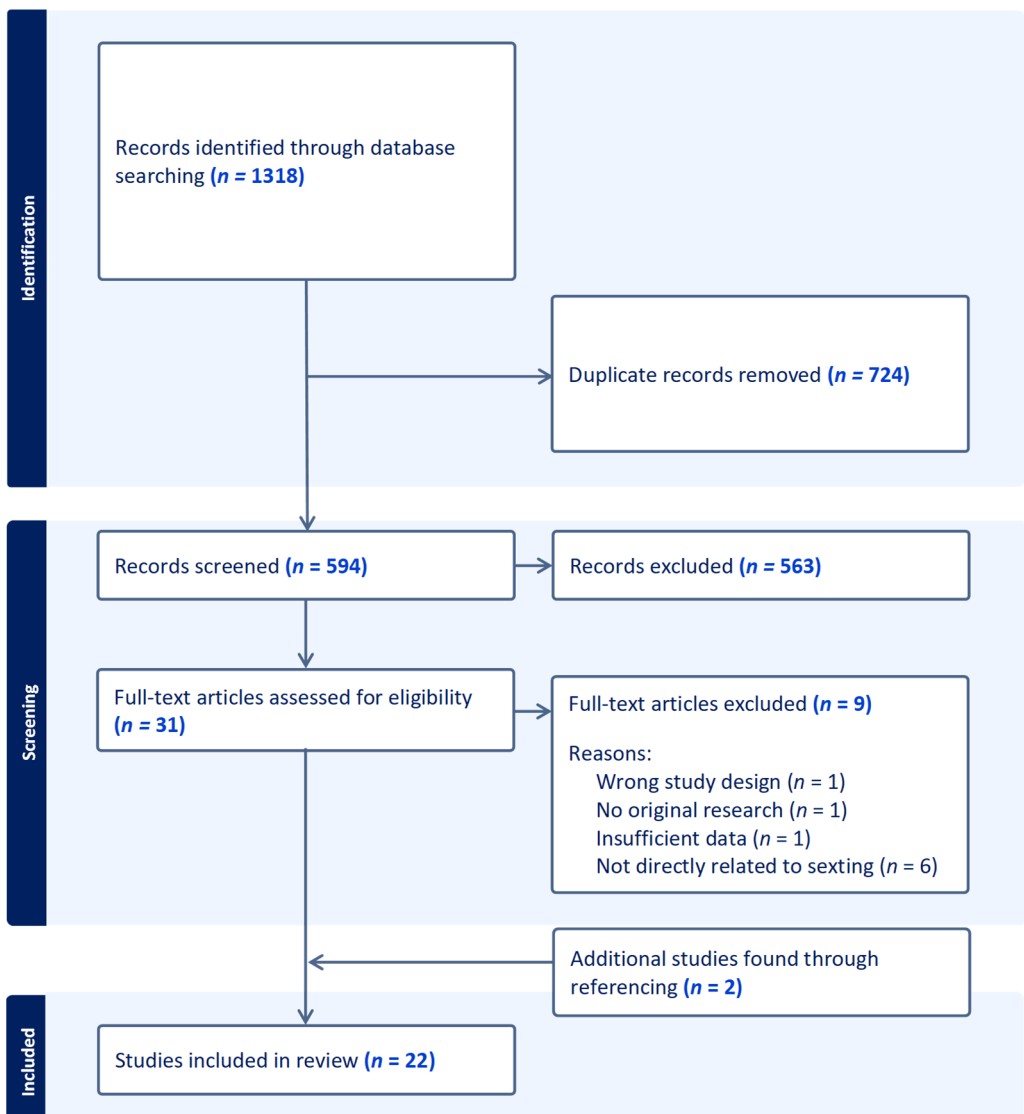

**Figure 1.** PRIMSA flowchart of the study selection process.

### 2.3. Quality Appraisal

Quality appraisal for the 24 studies included in the present review was conducted by two authors using the Joanna Briggs Institute (JBI) Critical Appraisal Checklist for Analytical Cross-Sectional Studies [33]. The checklist comprised 8 questions regarding study design: selection bias; descriptiveness of setting and sample; validity, reliability, and objectivity of measurements; presence of confounding factors; adjustments for key confounds; and appropriateness of statistical analysis. Response options included "yes", indicating high quality; "no" indicating poor quality; "unclear"; and "not applicable". Reviewers then provided an overall quality appraisal of the study, indicating whether the quality of the evidence was "Good"; "Fair"; or "Poor". Any discrepancies in judgment were resolved through discussion.

## 3. Results

Based on the JBI critical appraisal tool, the overall quality of studies was good ($n = 10$) and fair ($n = 14$); no eligible studies received a poor-quality rating. Item 4 on the checklist was not applicable to most studies, as most studies did not include a treatment condition or limit the sample to a particular mental-health condition.

### 3.1. Demographics and Definitions

Overall, 24 studies were included in the review, representing an analytical sample of 21,658 participants. Follow-up periods ranged from 2 months to 4 years. The majority of the studies were conducted using participants from the United States ($n = 9$), followed by Spain ($n = 5$) and the Netherlands ($n = 3$). Most studies recruited adolescent samples at baseline ($n = 20$), with only three studies [34–36] using young adult and/or adult samples throughout. Table 2 provides a summary of the included studies and relevant extracted data.

Sexting definitions varied across studies in regard to the type of sext (video, image, text), the method of exchange (sending, receiving, forwarding), and the presence of consent. Further, a small number of studies ($n = 5$) contextualized sexting as part of a suite of broader online behaviours. Two studies [37,38] conceptualized sext-sending as a form of risky sexual online behaviour, which also included online sexual solicitation, seeking sexual online communication, and divulging personal information online. One study [34] conceptualized pressured sexting as a part of cyber sexual aggression, which also included online sexual harassment, hassling, and virtual rape behaviours. Another study [35] presented sext-sending and sexy self-presentation (i.e., displaying oneself in an intimate manner on social media) combined as online sexual engagement. Finally, one study [39] measured sexually related online activities, which included sext-sending, sext receiving, engaging in virtual sex, and discussing sexual experiences with unknown individuals.

**Table 2.** Summary of Findings.

| Author | Country | Final Sample *n* (% Women) | Sampling | Age Range, *M*, *SD* (Years) | Follow-Up (Time Lag) | Sexting Definition | Key Variables | Findings |
|---|---|---|---|---|---|---|---|---|
| Alonso and Romero [14] | Spain | 624 (55.0%), cisgender | CS, High school students | 12–19, *M* = 14.35, *SD* = 1.55 [b] | 2 (12 months) | The sending of sexually explicit photos, videos and text messages. | Sexting, big five personality traits, bullying/cyberbullying, emotional wellbeing. | High extraversion, low agreeableness, low conscientiousness, and aspects of neuroticism (high vulnerability, impulsivity, depression) at baseline predicted sexting at follow-up. Sexting at baseline decreased bullying victimization and positive emotions at follow-up. However, sexting was longitudinally unrelated to cyberbullying and negative emotions. |
| Baumgartner et al. [37] | Netherlands | 1445 (49.0%), cisgender | RS | 12–17, *M* = 14.5, *SD* = 1.68 [a] | 2 (12 months) | Sent on the internet a photo or video on which they were partly naked to someone they knew only online. Contextualized as part of Risky Sexual Online Behaviour. | Risky sexual online behaviour, perceptions of peer involvement, risks, and benefits in risky sexual online Behaviour, | Perception of peer involvement in risky online behaviour at baseline was associated with risky online behaviour during follow-up. Perception of risks and benefits was not associated with risky online behaviour. |
| Baumgartner et al. [38] | Netherlands | 1016 (50.3%), cisgender | RS | 12–17, *M* = 14.5, *SD* = 1.68 [a] | 4 (6 months) | Sending a photo or video in which they were partly naked to someone they only knew online. Contextualized as part of Risky Sexual Online Behaviour. | Risky sexual online behaviour, descriptive and injunctive peer norms | Descriptive and injunctive peer norms at baseline predicted adolescents' engagement in risky sexual online behaviour during subsequent follow-up periods. |
| Bogner et al. [40] | USA | 343 (51.0%), cisgender | CS, Middle school students | 12–14, *M* = 12.89, n/a [b] | 2 (12 months) | Transmission of sexual pictures/messages via an electronic device. | Sexting, first-time offline sexual behaviour | Sexting at baseline predicted first-time oral and penetrative sex at follow-up |
| Brinkley et al. [41] | USA | 181 (46.9%), cisgender | CS, High school students | 15–16, n/a, n/a [a] | 2 (24 months) | Sending sexually explicit or suggestive. images, videos, or text messages via digital communication | Text-based sexting, offline sexual activity, drug use, borderline personality features. | Text-based sexting at baseline predicted sexual intercourse, drug use in conjunction with sexual intercourse, and multiple partners at follow-up. Discussion of hypothetical sex at baseline predicted borderline personality features at follow-up. |

**Table 2.** *Cont.*

| Author | Country | Final Sample *n* (% Women) | Sampling | Age Range, *M*, *SD* (Years) | Follow-Up (Time Lag) | Sexting Definition | Key Variables | Findings |
|---|---|---|---|---|---|---|---|---|
| Burić et al. [42] | Croatia | 859 (100%), cisgender | CS, Secondary school students | n/a, *M* = 15.8, *SD* = 0.50 [a] | 5 (4 months) | Sending and/or receiving sexually suggestive texts, photos, or videos typically of oneself. | Sexting, depression, anxiety, self-esteem, peer conformism, adverse family environment | No longitudinal relationship was found between sexting, depression, anxiety and self-esteem. Adverse family environment and peer conformism at baseline was related to more frequent sexting, lower levels of self-esteem, and higher levels of depression and anxiety during subsequent follow-up periods. |
| Casas et al. [43] | Spain | 1431 (46.4%), cisgender | CS, Secondary school students | 11–18, *M* = 13.61, *SD* = 1.31 [b] | 2 (4 months) | Sending receiving and forwarding sexually suggestive and explicit images, videos or text messages using a mobile phone, Internet and other electronic device. | Sexting, sexting normalization, social competence, need for popularity, willingness to sext, cyber gossip | Need for popularity, participation in cyber gossip, lower social competence, normalization of sexting and willingness to sext at baseline predicted involvement in sexting behaviours at follow-up. |
| Chang et al. [44] | Taiwan | 2315 (n/a) | CS, High school students | Year 10, n/a, n/a [a] | 2 (12 months) | Unwanted Sext Receiving: . . . open an email or instant message or a link in a message that showed you pictures of naked people or people having sex that you did not want. Presented as unwanted exposure to online pornography. | Unwanted exposure to online pornography, online sexual solicitation, pornography use, internet risk behaviour, cyberbullying, offline sexual harassment, internet use | Pornography use, internet risk behaviour, and cyberbullying experience at baseline predicted unwanted exposure to online pornography at follow-up (i.e., received unwanted sexts). |
| Chaudhary et al. [45] | USA | 500 (52.4%), cisgender | CS, Middle school students | n/a, *M* = 12.22, *SD* = 0.57 [b] | 2 (12 months) | Sending or posting sexually suggestive text messages, videos and images, including nude or semi-nude photographs or videos, via cellular telephones or over the Internet (such as email or social networking site like Facebook). | Sexting, anxiety, depression | Sexting at baseline was significantly associated with anxiety and depression during follow-up. |

**Table 2.** *Cont.*

| Author | Country | Final Sample *n* (% Women) | Sampling | Age Range, *M*, *SD* (Years) | Follow-Up (Time Lag) | Sexting Definition | Key Variables | Findings |
|---|---|---|---|---|---|---|---|---|
| Choi et al. [46] | USA | 758 (60.7%), cisgender | CS, High school students | n/a, *M* = 20.03, *SD* = 0.76 [b] | 4 (12 months) | The exchange of self made sexually explicit photographs via digital media. | Sexting, number of sexual partners, offline sexual behaviour | Sexual activity and number of sexual partners were positively associated with sexting at baseline and subsequent follow-up periods No longitudinal relationships were identified. |
| Daskaluk [34] | Canada | 143 (100%) | CS, Undergraduate students | 18–25, *M* = 20.1, n/a | 2 (4 months) | Pressured Sexting: Pressured you to send a sexual or naked photo/video of yourself. Contextualized as part of cyber sexual aggression. | Cyber sexual aggression, depression, anxiety, stress, sexual satisfaction, relationship satisfaction, self esteem | No longitudinal associations emerged between cyber sexual aggression and depression, anxiety, stress, sexual satisfaction, relationship satisfaction, or self-esteem. Cyber sexual aggression was associated with relationship satisfaction at baseline. |
| Dodaj et al. [47] | Croatia | 359 (60.2%), cisgender | CS, High school students | 15–17, 16.32(0.64) [b] | 2 (12 months) | Exchange of sexually explicit or provocative contents (text messages, photographs, and video recordings) using a mobile phone, the internet, or social networks. | Sexting, depression, anxiety, stress | No longitudinal relationships between psychological distress (depression, anxiety, stress) and sexting behaviour was identified. Stress was a predictor of sext-sending at baseline, while depression was a significant predictor of sext receiving and sending at follow-up. |
| Gámez-Guadix and de Santisteban, [48] | Spain | 1208 (52.8%), cisgender | CS, Secondary school students | 12–16, *M* = 13.57, *SD* = 1.09 [b] | 2 (12 months) | The production and sending of sexual content via the Internet and smartphones. | Sexting, Big 5 personality traits, depression, self-esteem and problematic internet use. | Lower conscientiousness, higher extraversion, and higher depression at baseline predicted sext-sending at follow-up. No longitudinal relationship between self-esteem, problematic internet use and sexting was found. |
| Gámez-Guadix and Mateos-Pérez [49] | Spain | 1497 (53.2%), cisgender | CS, Secondary school students | 12–14, *M* = 13.65, *SD* = 0.79 [b] | 2 (12 months) | The creation and exchange of text messages, photos, or videos with sexual or erotic content via the Internet or smartphones. | Sexting, sexual solicitations, cyberbullying | Sexting at baseline predicted receiving sexual solicitations by adults and cyberbullying victimization at follow-up.Receiving sexual solicitations by adults and cyberbullying at baseline predicted sexting behaviour at follow-up. |

**Table 2.** *Cont.*

| Author | Country | Final Sample n (% Women) | Sampling | Age Range, *M*, *SD* (Years) | Follow-Up (Time Lag) | Sexting Definition | Key Variables | Findings |
|---|---|---|---|---|---|---|---|---|
| Hicks et al. [50] | USA | 887 (42%), cisgender | CS, Secondary school students | 11–17, n/a, n/a | 4 (12 months) | . . . ever sent sexual messages or pictures to someone you had romantic or flirtatious experiences with, whether or not you were dating. | Sexting, offline sexual behaviour, race | Sexting at baseline1 predicted oral and sexual intercourse at follow-up for black males, white males, and white females but not for black females. oral sex and sexual intercourse at baseline did not predict sexting during subsequent follow-up periods. |
| Kurup et al. [51] | USA | 197 (48%), cisgender | CS, High school students, Text messages | n/a, *M* = 13.93, n/a [a] | 4 (12 months) | Text-based messages about sexual behaviours that occurred within dyad (i.e., involving the sender and receiver rather than anyone outside the conversation). | Sexting, internalizing, externalizing, and social problems; borderline personality features, life satisfaction, group belongingness, positive self-perceptions | No longitudinal relationships were found. Text-based sexting was cross-sectionally associated with lower levels of group belongingness. For women, text-based sexting was related to increased externalizing symptoms and borderline personality disorder features, as well as decreased life satisfaction, group belongingness, self-perceived social competence, and global self-worth. |
| Ojeda et al. [52] | Spain | 1736 (46.3%), cisgender | CS, High school students | 12–16, *M* = 13.60, *SD* = 1.25 [b] | 2 (4 months) | Sending, receiving, and forwarding of sexually suggestive and explicit images, videos or text messages through mobile phones, the internet, and other electronic media. | Sexting, bullying perpetration, cyberbullying perpetration | Cyberbullying at baseline was not associated with engagement in any subsequent sexting. Sexting at baseline was not associated with later cyberbullying or bullying. Bullying behaviours at baseline predicted third-party forwarding of sexual content at follow-up. |
| Reer et al. [35] | Germany | 586 (54.4%), cisgender | CS, Online Panel | 14–64, *M* = 41.30, *SD* = 13.91 [a] | 2 (12 months) | Sharing of personal, sexually suggestive text messages, or nude or nearly nude photographs or videos via electronic devices. Contextualized as part of online sexual engagement. | Online sexual engagement, online sexual victimization, loneliness, life satisfaction, depression, anxiety. | Online sexual engagement at baseline was positively associated with online sexual victimization at follow-up. No longitudinal associations between psychosocial wellbeing and sexting were identified. |

**Table 2.** *Cont.*

| Author | Country | Final Sample *n* (% Women) | Sampling | Age Range, *M*, *SD* (Years) | Follow-Up (Time Lag) | Sexting Definition | Key Variables | Findings |
|---|---|---|---|---|---|---|---|---|
| Ševčíková et al. [53] | Czech Republic | 323 (51.1%), cisgender | n/a, Research Panel | 15, n/a, n/a [a] | 2 (24 months) | Received erotic photos from somebody; sent their own erotic photos to somebody Contextualized as part of sexually related online activities. | Sexually related online activities, peer attachment, pubertal status, offline sexual behaviour | Poor peer attachment, advanced pubertal status, and prior offline sexual experiences in adolescents at baseline predicted sexually related online activities at follow-up |
| Ševčíková et al. [39] | Czech Republic | 1134 (58.8%), cisgender | CS, Primary and Secondary School Students | 10–18, *M* = 13.84, *SD* = 1.94 [b] | 3 (6 months) | In the last 6 months, I sent an erotic photo of me on the internet to my girlfriend/boyfriend. | Sexting, sensation seeking, offline sexual behaviour | Sexting at baseline predicted offline sexual behaviour at follow-up. No longitudinal relationship between sensation seeking and sexting were found. |
| Steinberg et al. [54] | USA | 429 (54%), cisgender | CS, High school students | Year 9, n/a, n/a [a] | 3 (12 months) | Sending sexually explicit messages, images, or videos to a romantic partner. | Sexting, Offline Sexual Behaviour | Sexting co-emerges with offline sexual behaviour. No longitudinal relationships were reported. |
| Temple and Choi [46] | USA | 964 (56%), cisgender | CS, High school students | n/a, *M* = 16.09, *SD* = 0.79 [b] | 2 (12 months) | Electronically sending sexually explicit images from 1 adolescent to another. | Offline sexual behaviour, risky offline sexual behaviour | Sexting at baseline predicted offline sexual activity at follow-up. No longitudinal relationship emerged between sexting and risky sexual behaviours. |
| van Oosten and Vandenbosch [36] | Netherlands | 1947 (51.8%), cisgender | RS | 13–25, n/a, n/a | 2 (2 months) | Non-consensual forwarding of sexts was defined as: forwarding a sexually explicit picture (or video) of someone without that person's consent. | Willingness to non-consensually forward sexts, online pornography use, instrumental attitudes to sext. | Pornography use at baseline predicted a higher willingness to non-consensually forward sexts during follow-up, but mostly among adolescent boys (aged 13–17) with high levels of instrumental attitudes towards sex. |
| Van Ouytsel et al. [55] | USA | 776 (57.6%), cisgender | CS, High school students | 13–18, *M* = 16.1, *SD* = 0.79 [a] | 3 (12 months) | The sending or receiving of sexually explicit messages, images, or videos through the internet or mobile phone. | Sexting, cyberbullying victimization, bullying victimization | Sexting at baseline was associated with subsequent cyberbullying victimization across all subsequent follow-up periods. Sexting at second follow-up was also associated with bullying victimization at the final follow-up period. Cyberbullying victimization at second follow-up was associated with sexting at final follow-up. |

Note: n/a = information not reported, [a] = at Baseline, [b] = at Final Follow-up, CS = convenience sample, RS = nationally representative sample.

### 3.2. Cyberbullying/Bullying

In total, five studies investigated longitudinal relationships between sexting and cyberbullying/bullying behaviour, all conducted with adolescent samples. Three of these studies [14,49,56] explored longitudinal associations between sext-sending, and cyberbullying/bullying victimization, while one study [44] investigated the relationship between receiving unwanted sexts and cyberbullying victimization, and, lastly, one study [52] explored sexting engagement and cyberbullying/bullying perpetration.

Alonso and Romero [14] found that sext-sending at baseline predicted less bullying victimization during follow-up but was unrelated to cyberbullying victimization. The authors suggested that given sexting was more common in socially popular adolescents, those who sexted were also less likely to experience bullying victimization [14]. In contrast, Gámez-Guadix and Mateos-Pérez [49] found that sexting at baseline predicted future cyberbullying victimization, and cyberbullying victimization at baseline was predictive of future sexting behaviours, suggesting a bidirectional relationship [49]. Along similar lines, Van Ouytsel et al. [56] found a strong bidirectional relationship between sext-sending and cyberbullying/bullying victimization. Specifically, initial sext-sending predicted cyberbullying and bullying victimization at later time points, while cyberbullying victimization at earlier time points predicted subsequent sext-sending [56].

Further, Chang et al. [44] found that cyberbullying victimization and perpetration at baseline were associated with the subsequent receiving of unwanted sexts, presented in this study as unwanted exposure to online pornography [44]. Lastly, Ojeda et al. [52] found cyberbullying perpetration at baseline was not associated with subsequent sext-sending. Similarly, initial sext-sending was not associated with cyberbullying/bullying perpetration behaviours during follow-up. However, the longitudinal analysis found that bullying perpetration at baseline was associated with subsequent third-party forwarding of sexual content. Overall, the existing literature on the causal relationship between sexting and cyberbullying/bullying experiences is limited and inconclusive, with studies focused on different facets of behaviour (e.g., victimization, perpetration).

### 3.3. Offline Sexual Behaviour

Eight longitudinal studies investigated associations between sexting and offline sexual behaviour [39–41,46,50,53,54,57], all based on adolescent samples. Overall, reviewed studies found a positive association between sexting and engagement in offline sexual behaviours over time. Five studies found sext-sending behaviours at baseline predicted future engagement in offline sexual behaviours, including oral and vaginal intercourse [39–41,50,57]. Bogner et al. [40] found that initial sext-sending in participants with no history of oral or penetrative sex was predictive of first-time sexual contact during follow-up. Hicks et al. [50] expanded on this finding, delineating by racial background, and found that sexting was associated with offline sexual contact for white women and men and black men, but not black women. The authors attributed this finding to the suppressive discourse restricting black women's sexual agency. Further, Brinkley et al. [41] identified text-based sexting as a predictor of multiple sexual partners and engagement in drug use in combination with offline sexual activity. Finally, Temple and Choi et al. [46] found a significant association between sexting at baseline and future offline sexual behaviour. Interestingly, the study reported no longitudinal relationship between sexting at baseline and future risky offline sexual behaviours.

Findings on the predictive effect of offline-sexual behaviours at baseline on sexting behaviour during follow-up were mixed. Hicks et al. [50] found no association between offline sexual behaviours at baseline and subsequent sexting, while Ševčíková et al. [53]. found offline sexual experiences predicted later sexting behaviours. It should be noted that Ševčíková et al. [53] explored sexting as part of sexually related online activity, which also included engaging in virtual sex and discussing sexual experiences online with strangers. This variance in definitions may explain the difference in findings between these studies. Finally, two studies [46,54] found sexting to co-occur with offline sexual behaviour, though

no longitudinal associations were reported. Overall, while the causal relationship between initial sext-sending and subsequent offline-sexual behaviour is well established in current research; evidence regarding the predictive effect of offline-sexual behaviour on sext-sending is less clear.

### 3.4. Mental Health

Overall, eight studies provided longitudinal findings regarding associations between sexting and psychological outcomes, with mixed findings. Four studies found sexting behaviours at baseline did not predict future symptoms of depression and anxiety [34,35,42,47]. In contrast, one study found that sexting at baseline predicted symptoms of anxiety and depression during follow-up [45]. Two studies [14,48] found that depression at baseline predicted subsequent sext-sending behaviours.

Alonso and Romero [14] found that sext-sending at baseline was predictive of a decrease in positive emotions (e.g., excitement, pride) experienced at follow-up. Addressing the relationship between sexting and self-esteem, studies found no longitudinal relationship [34,42,48]. Finally, one study [51] utilized observational methods to investigate text-based sexting and psychosocial adjustment. Kurup et al., [51] found no longitudinal associations but reported a cross-sectional association between text-based sexting and lower group belongingness.

It is worth noting that while the majority of the above studies (*n* = 6) examined mental health in adolescents, Daskaluk [34] examined sexting behaviours in young adults aged 18–25, while Reer et al. [35] focused on behaviours across a broad age range (14–61). Further, one of these studies [34] examined pressured sexting as a part of cyber sexual aggression, which included online sexual harassment, hassling, and virtual rape. It is important to note that although the two aforementioned studies had varying conceptualizations of sexting behaviours, findings are broadly consistent. In summary, longitudinal findings on the causal association between sexting and mental health outcomes were mixed, with studies reporting either no effect or a negative impact on mental health.

### 3.5. Social Norms

In total, five studies investigated predictive relationships between social norms and sexting behaviour [37,38,42,43,53], all based on adolescent samples. Two studies investigated the impact of subjective beliefs about peer behaviour (i.e., peer norms) on risky online behaviour, which included the following behaviours: sexting, online sexual solicitation, online sexual communication, and divulging personal information online [37,38]. One of these studies found that the perception of peer involvement in risky online behaviour predicted engagement in risky online behaviour at follow-up. However, the perception of the risks and benefits of risky online behaviour had no such impact [37]. Additionally, Baumgartner et al. [38] found that belief in positive peer norms regarding risky online behaviour predicted future engagement in risky online behaviour. Along similar lines, peer conformism was found to be causally related to more frequent sext-sending [42]; Burić et al. [42] also found that adverse family environment at baseline was predictive of sext-sending during follow-up.

The results of these three studies [37,38,42] suggest a causal relationship between perceived positive peer norms regarding sexting and engagement in sext-sending. This is further supported by Casas et al. [43], who found a causal link between the endorsement of normal or usual attitudes towards sexting (e.g., "sending erotic/sexual videos, photos, or messages is normal, it's fine") and sext-sending. The study also identified a number of social traits that predicted sext-sending during follow-up, including a greater need for popularity, engagement in cyber gossip, lower social competence, and the willingness to sext [43]. Finally, Ševčíková et al. [53] found poor peer attachment at baseline predicted subsequent engagement in sexually related online activities, including sext-sending. In general, studies found that normalized sext-sending attitudes, positive peer norms towards sexting, and a desire to conform to such norms predicted future engagement in sexting.

*3.6. Additional Factors*

A number of additional factors were explored in additional longitudinal studies on sexting. Given the relatively low number of studies investigating these factors, they are discussed generally. Two studies investigated personality traits [14,48], two studies investigated pornography use [36,44], and one study investigated sexual solicitation [49]. Although three other studies also included sexual solicitation as part of their research [37,38,44], two of these studies [37,38] consolidated sexual solicitation and sexting into a third variable (i.e., risky sexual online behaviour), while the other study did not analyse sexual solicitation in relation to sexting [44]. Therefore, the aforementioned three studies could not be included in this subsequent analysis.

Investigation of relationships between personality traits and sexting found that high levels of extraversion and lower reported levels of conscientiousness at baseline predicted future sext-sending behaviour [14,48]. One study also found that low agreeableness at baseline predicted sext-sending [14]. However, a second study [48] found no such association. Finally, one study found texting about hypothetical sex to be predictive of borderline personality features at a later time, even when controlling for borderline personality features at baseline [41].

Studies investigating pornography use found that exposure to pornography in adolescents predicted the receiving of unwanted sexts [44]. Further, pornography use at baseline predicted a greater willingness to non-consensually forward sexts among both young adults and adolescents. This relationship was especially strong among adolescent boys with high instrumental attitudes towards sex [36]. Findings from these two studies suggest a close relationship between pornography and harmful sexting behaviours [36,44]. Finally, a bidirectional causal relationship was reported between the receiving of sexual solicitations from adults and sext-sending in adolescents [49].

## 4. Discussion

Research on sexting is no longer in its infancy, given the number of existing studies in multiple disciplines and synthesis in existing reviews [2,5–7,26]. However, empirical literature that identifies causal relationships between sexting and other associated factors is lacking, and currently, no reviews exist on prior longitudinal research into sexting. The areas of focus in existing longitudinal studies remain diverse, and at the time of this review, no comprehensive longitudinal review had been undertaken to identify causal patterns in sexting behaviours. The current study provides a review of longitudinal research on sexting behaviour by synthesizing disparate literature and highlighting trends in the findings.

This review identified a diverse range of studies, with notable variations in definitions of sexting and the associated factors being examined. The breadth of approach may be reflective of variations in the disciplinary backgrounds or traditions of researchers. For example, researchers based on health and developmental psychology traditions may take a life-course approach, considering associations with sexual behaviour, mental health, peer norms, and group belongingness [35,40,51]. In contrast, other researchers were predominantly focused on risky and/or antisocial associations, e.g., bullying and cyberbullying, online sexual solicitation, and harassment [36,44,51].

There was significant variety in the directionality associated with sexting behaviours, whereby some studies measured variables as potential predictors of sexting [49], while others measured the same variables as consequences of sexting behaviours [14]. Further, few studies differentiated between consensual and non-consensual sexting. This distinction is critical, especially for studies investigating the potential for harmful consequences due to sexting, as consent has been found to be a moderating factor when considering harmful outcomes of sexting [20,26,27,48]. Such varied definitions and conceptualizations reflect the broad scope of the existing research, which may contribute to the difficulty in finding consensus across the sexting literature and explain why this review identified variations across a range of outcomes.

Although cross-sectional research has suggested a strong association between sexting and cyberbullying/bullying victimization [9,12,58,59], longitudinal findings are more mixed. Two studies [49,55] provided evidence of a significant association between sext-sending and cyberbullying victimization, while one study found no such significant relationship [14]. Notably, conflicting results were found by Alonso and Romero [14] and Van Ouytsel et al. [56] in regard to the association between sexting and offline bullying victimization. One study [14] suggested that sexting was more commonly undertaken by popular adolescents and therefore could lead to lower instances of bullying victimization. On the other hand, Van Ouytsel et al. [56] suggested that sexting may lead to risky outcomes such as the non-consensual sharing of intimate images, which may, in turn, lead to bullying victimization.

Two additional studies [44,52] suggested that a causal relationship may exist between cyberbullying perpetration and unwanted/non-consensual sexting behaviours. Chang et al. [44] found both cyberbullying perpetration and victimization to be associated with unwanted sext receiving. Similarly, Ojeda et al. [40] found cyberbullying perpetration to be associated with the forwarding of sexts, a sexting behaviour that is primarily, but not exclusively, engaged in without consent [60,61]. As such, it appears that issues of consent may be critical in determining cyberbullying outcomes associated with sexting.

Unsurprisingly, all seven relevant studies found significant associations between sexting and offline sexual behaviours. Two of these studies, whilst longitudinal, performed only cross-sectional analyses [46,54], while five studies [39–41,50,57] found that sexting behaviours predicted sexual intercourse at a later time. These findings suggest that sexting may act as a gateway behaviour to offline sexual interactions.

However, studies on offline sexual behaviour as a predictor for future sexting engagement produced mixed results. According to Ševčíková et al. [53], offline sexual intercourse predicts future engagement in sexting behaviour, while another study found that no such association exists [50]. It should be noted that Ševčíková et al. [53] acknowledged a lack of data on sexting behaviours at baseline, which prevents any conclusion on whether sexting is a consequence of offline sexual intercourse or vice versa. However, evidence from Hicks et al. [56] indicates that sexting may act as a precursor to offline sexual behaviour. However, the evidence suggests that offline sexual behaviour may be less strongly associated with future online sexual interactions such as sexting, which are perhaps perceived as riskier.

A previous systematic review regarding sexting behaviours [2] found that 12 out of 14 observed studies suggested relationships between sexting and depressive symptoms. However, the present review found mixed longitudinal evidence regarding relationships between sexting and psychological outcomes. While three studies found a longitudinal relationship between sexting and symptoms of depression and anxiety [14,45,48], four studies found no such association [34,35,42,47]. It should be noted that Daskaluk [34] examined pressured sexting as part of a broader set of online aggressive behaviours (e.g., sexual harassment, hassling, and virtual rape behaviours), thus using a broader operationalization.

An additional potential explanation for differences in findings may lie in the ages of the participants. Specifically, the average age for participants in studies that found no longitudinal associations between sexting and symptoms of depression/anxiety was older (*M age* = 23.38) than those in studies that did (*M age* = 13.38). Therefore, these longitudinal findings may reflect the notion that the psychological risk factors and outcomes typically associated with sexting may be moderated by the victim's age [59].

Research investigating the association between self-esteem and subsequent sexting found no longitudinal relationship between these variables [34,42,48]. Therefore, longitudinal evidence does not support the notion that teens sext as a result of low self-esteem. Finally, one study investigated the association between text-based sexting and a number of psychosocial consequences and found no longitudinal relationship [51].

Considering peer and social factors, findings suggest that factors associated with the normalization of sexting behaviours predicted future engagement in said behaviour [43].

The belief that one's peers are involved in and hold favourable views regarding sexting predicted future sext-sending behaviour [37,38]. Further, being alienated from peers was also found to predict subsequent engagement in sexting behaviours [53]. It is possible that sexting behaviours are used as a method to compensate for poor relations with peers, which is consistent with findings that high levels of peer conformism were associated with subsequent sexting behaviours [42]. These results are consistent with previous reviews which found that sexting behaviours are strongly influenced by peer endorsement and peer pressure [2,5,7].

Further, findings suggested that high extraversion and lower conscientiousness predicted future engagement in sexting [14,48]. Alonso and Romero [14] posit that individuals with higher extraversion may possess a need for more "thrilling experiences", which may be provided via sexting engagement, while individuals with lower conscientiousness are likely to possess lower inhibitions and are therefore more prone to sext. Consistent with this, a recent cross-sectional, large sample study ($n$ = 5542) on the association between personality traits and sexting across adolescents and adults found that high extraversion and lower conscientiousness were associated with not only sext-sending but also risky and aggravated sexting behaviour [62]. Finally, texting about hypothetical sex may also facilitate impulsive, sensation-seeking behaviours reflective in borderline personality [54].

In our review, studies exploring the relationship with pornography use found a relationship between pornography viewing and the unwanted/non-consensual forms of sexting behaviours. For example, Chang et al. [44] found that pornography use predicted a greater likelihood of receiving unwanted sexts, though they did not consider other sexting behaviours. Further, van Oosten and Vandenbosch [36] suggest that pornography use predicted a greater willingness to share intimate sexts without consent in both adolescents and young adults.

The normalization of sexual content distribution may in turn encourage one to share intimate images themselves. This is supported by prior studies of motivations for non-consensual sext sharing [19,60,61]. Given the paucity of studies, additional research is needed in order to draw conclusions on whether pornography use is a risk factor for engagement in non-consensual sexting behaviours. Lastly, Gámez-Guadix and Mateos-Pérez [49] noted bidirectional relationships between adolescent sext-sending and the receiving of sexual solicitations from adults. It is possible that online sexual victimization can increase the likelihood of adolescents engaging in sexual behaviours online. This, in turn, can increase exposure to perpetrators [49], thus fuelling a problematic cycle of online victimization.

### 4.1. Implications

Findings from this review offer a number of implications for researchers. Importantly, it is critical for future research to delineate between consensual sexting behaviours and non-consensual, aggravated online behaviours such as non-consensual sext dissemination and online sexual harassment. This distinction is critical to provide greater clarity to measurements and future comparison across studies, provided that reviewed studies have indicated that sexting is a precursor to offline sexual behaviour and is likely a normative part of sexual expression in relationships. Those who are involved with adolescents should instead adopt a normalcy discourse while discussing the topic, approach sexting with harm-minimization in mind, and shift prevention effects away from abstinence-based sex education [63]. More research on help-seeking behaviours and how people may protect themselves while sexting is important and may inform support programs for potential victims.

However, studies have also shown that sexting is not a purely benign behaviour and may lead to a number of harmful consequences such as cyberbullying and bullying victimization and unwanted sexual solicitation. A better understanding of the association between harmful subsets of sexting behaviour, such as non-consensual sext dissemination and pressured sexting, and the aforementioned consequences is needed.

In addition, children and adolescents seem to be particularly vulnerable to the harmful consequences of sexting, with younger persons more likely to experience depressive symptoms as a result of these behaviours. It is possible that the assessment of whether sexting is normative or harmful is moderated by age. Relevantly, there is a lack of longitudinal research exploring sexting behaviours in young adults/adults. Future research could add to this area of knowledge, which, in turn, would inform where prevention efforts should be targeted.

Finally, high extraversion, low conscientiousness, and the willingness to comply with prevailing peer pressure and norms were found to be causally associated with sexting behaviour. These findings contribute to the development of education and prevention programs targeted towards harmful sexting behaviours. Future programs should emphasize how best to resist peer pressure and address normative expectations around harmful sexting behaviour among youth. Further, findings suggest that prevention strategies would benefit from a focus on adolescents high in extraversion and low in conscientiousness.

*4.2. Limitations*

Several limitations to this review have been identified in order to provide context for the results. First, this review is limited to published works and may be subject to publication bias. Second, the variance in definitions, measured variables, age differences, time lag between baseline and follow-up, or rate of attrition may confound the conclusions of this review. Thirdly, this review is limited to English publications, and thus results cannot be generalized to studies published in other languages. Fourth, the majority of studies included in this review ($n$ = 22) utilized self-report methods of collecting data, which may have introduced bias to the results of the studies themselves, and focused largely on adolescent or at most early adult populations, hence there is limited data regarding sexting behaviours. Fifthly, due to high levels of heterogeneity, meta-analytic review techniques were not appropriate, and hence, this review presents conceptual findings but cannot comment on statistical findings. Finally, this review is limited to the databases and search terms presented in the methods section (Section 2). Studies that did not include the keywords utilized in our review would not have been included.

**5. Conclusions**

Overall, longitudinal research into sexting remains scarce. Despite the range of search terms utilized, only 24 studies were identified for analysis in this study. Identified papers varied greatly in how sexting was contextualized and in which specific variables were examined, reflecting the broad range of disciplines investigating sexting behaviours. The variability in included studies created a number of challenges in synthesizing findings. However, despite the lack of consensus among longitudinal results, this review presents a number of implications for future research, as well as support for previous findings in cross-sectional research.

This review found that harmful behaviours such as bullying and cyberbullying, as well as psychological outcomes such as depression and anxiety, were not consistently predictive of sexting behaviours, as findings have been largely mixed. In contrast, findings on the predictive effect of bullying and pornography viewing on non-consensual or unwanted sexting behaviours are largely consistent. These findings are relevant to future research as they, again, point to the importance of differentiating between consensual and non-consensual sexting behaviours.

Along similar lines, findings suggest that general sexting behaviours may be considered harmful behaviour for younger adolescents, possibly because they have difficulty sufficiently understanding the scope of such behaviours to provide informed consent. One major area of consensus among existing longitudinal sexting research is the association between sexting and offline sexual activity, signalling that sexting may be a method of sexual experimentation and a precursor to a broader set of sexual behaviours. Perceived peer norms surrounding sexual behaviour have also emerged as a causal factor, which

seems to contextualize sexting as normalized behaviour, driven by peer endorsement and pressure. It is possible that future prevention efforts in schools may benefit from utilizing this information in mitigating the harmful outcomes of sexting.

**Supplementary Materials:** The following supporting information can be downloaded at: https://www.mdpi.com/article/10.3390/psych5020035/s1, Figure S1: PRISMA 2020 Checklist; Table S1: Quality assessment using the Joanna Briggs Institute (JBI) Critical Appraisal Checklist for Analytical Cross-Sectional Studies.

**Author Contributions:** Conceptualization, B.K. and Y.H.; methodology, Y.H.; software, Y.H.; validation, Y.H., E.M.C. and B.K.; formal analysis, Y.H.; investigation, Y.H., B.K. and E.M.C.; resources, B.K.; data curation, Y.H. and B.K.; writing—original draft preparation, Y.H.; writing—review and editing, Y.H., E.M.C. and B.K.; visualization, Y.H.; supervision, B.K.; project administration. B.K. and E.M.C. All authors have read and agreed to the published version of the manuscript.

**Funding:** This research received no external funding.

**Institutional Review Board Statement:** As this was a secondary review of published papers, no institutional review was required. Due to rapid timeframes, this review was not registered, hence no registration and protocol is available.

**Informed Consent Statement:** Not applicable.

**Data Availability Statement:** All included studies are available from the primary authors. No analytic code was developed.

**Conflicts of Interest:** The authors declare no conflict of interest.

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
