# Peer review of "Longitudinal Sexting Research: A Systematic Review"

_psych, doi:10.3390/psych5020035_

Round 1
Reviewer 1 Report
ABSTRACT: Clear and concise.
1. INTRODUCTION: Good job synthesizing previous sexting literature and identifying a need for this study.
2. MATERIALS & METHODS: Length of study should have been a consideration for selection. Comparing a study with a follow up of two months vs. four years can yield much different results.
3. RESULTS: Excellent chart information, but the spacing issues throughout were distracting and need fixed.
4. DISCUSSION: Good job explaining the variations in results but would like more discussion in why these variations between studies may exist. Under implications, line 391 says “It is therefore necessary to shift prevention efforts away from abstinence-based approaches and towards education on engagement in healthy sexting.” While I understand the basis of this suggestion, what makes this approach complicated is that minors’ sexting is still illegal in some countries and can be attached to possible child pornography charges.
5. CONCLUSIONS:
Other comments: Omit all forms of “it” and replace with more specific word.
Author Response
ABSTRACT: Clear and concise.
- INTRODUCTION: Good job synthesizing previous sexting literature and identifying a need for this study.
- MATERIALS & METHODS: Length of study should have been a consideration for selection. Comparing a study with a follow up of two months vs. four years can yield much different results.
Thank you for highlighting this. The authors acknowledge that the overall length of study (and follow-up time) may impact results in longitudinal research. However, given the overall lack of research in the area, an exclusion criterion for “length of study” was not included. A sentence clarifying this position has been added under section 2.1. Study Selection Criteria. This issue was further listed as a limitation of the current review under section 4.2. Limitations.
- RESULTS: Excellent chart information, but the spacing issues throughout were distracting and need fixed.
Thank you for identifying this issue. Spacing throughout the results table has been reviewed and addressed.
- DISCUSSION: Good job explaining the variations in results but would like more discussion in why these variations between studies may exist. Under implications, line 391 says “It is therefore necessary to shift prevention efforts away from abstinence-based approaches and towards education on engagement in healthy sexting.” While I understand the basis of this suggestion, what makes this approach complicated is that minors’ sexting is still illegal in some countries and can be attached to possible child pornography charges.
Thank you for these suggestions. We have added additional comments regarding the potential reasons for variations between studies. We also acknowledge the legal ramifications of sexting for minors and have modified the recommendations under section 4.1 Implications to be more cautious and focused on normalcy discourse and harm minimization, rather than a simple departure from abstinence-based approach.
- CONCLUSIONS:
Other comments: Omit all forms of “it” and replace with more specific word
Thank you for this recommendation. The pronoun “it” (i.e., made in reference to a specific word) has been replaced where possible for better clarity. However, the anticipatory “it” (e,g., it is possible…) has not been altered, as this was not made in reference to any specific word.
Reviewer 2 Report
I would like to thank the authors and the Editorial Board for the opportunity to review the article submitted to MDPI’s Psych. The authors' manuscript refers to a very important topic: sexting.
Being a cybersexuality researcher myself, I’ve found the authors’ paper very interesting. I’m glad that in the Introduction section, the authors included both possible positive and negative consequences of sexting, and opted out of introducing it only in the negative view. I believe that some minor changes can increase the quality of the presented manuscript. I highly suggest that the authors rephrase/specify/add the following:
· Please elaborate on why the authors’ performed a literature review instead of a meta-analysis (e.g. because it was impossible due to the different operationalizations of sexting and related variables). The authors write “Fifthly, due to high levels of heterogeneity, meta-analytic review techniques were not appropriate and hence this review presents conceptual findings, but cannot comment on statistical findings.” This sentence is not correct. The meta-analytic approach was appropriate, but it was impossible due to various reasons.
· Please explain why the authors opted out of including sexting synonyms in their search strategy. Many sexting studies are hidden behind similar or broader terms, such as cybersex, cybersexuality, netsex, online sex, distance sex, etc. I believe that the authors’ approach with the usage of only those keywords presented in Table 1 could largely limit the possible results they could get. What was the methodology behind selecting the used keywords?
· The authors write “A random sample of 10% was reviewed by the second author for verification purposes.” If that approach is in line with current recommendations, please refer to them with a proper literature reference.
· In 3.4 authors refer to psychological well-being. It is a widely used term in regard to Carol Ryff’s psychological well-being theory. In 3.4 the authors refer to psychological mental health outcomes such as depression or anxiety, which are not a part of a PWB construct. What is more, in 3.4 authors refer to positive affective states, which are a part of a different theory, the subjective well-being model by Diener. I highly recommend that authors rephrase all PWB-and SWB- related terms into mental health-related ones because, in the current form, the presented paragraph is misleading and not in line with the widely used two theoretical models.
· The authors write “Findings from this review offer a number of implications for researchers and professionals developing education and prevention programs”. As a clinical practitioner and researcher myself, I cannot agree with this sentence. The presented literature search does not provide any guidelines for psychoeducational programs, what is more, it makes it even more difficult because the presented results are not in-line with each other. In its current form, the aforementioned authors’ sentence feels like an obligatory sentence which is present in many manuscripts but does not bring anything to the subject. If the authors believe that the presented results can help in psychoeducational program development, please elaborate and give proper examples of how.
· The authors write that “it is, therefore, necessary to shift prevention efforts away from abstinence-based approaches and towards education on engagement in healthy sexting”. On what bases, do the authors believe that abstinence-based approaches are commonly used? Please refer to a proper scientific publication. In past years, abstinence-based approaches are being waived towards a new and more effective control use model. The whole Implications section feels like it was written by a researcher and scientist rather than a practitioner. I highly recommend that authors back up their examples with examples and practical recommendations because, in the current form, it provides directions for future research.
Author Response
I would like to thank the authors and the Editorial Board for the opportunity to review the article submitted to MDPI’s Psych. The authors' manuscript refers to a very important topic: sexting.
Being a cybersexuality researcher myself, I’ve found the authors’ paper very interesting. I’m glad that in the Introduction section, the authors included both possible positive and negative consequences of sexting, and opted out of introducing it only in the negative view. I believe that some minor changes can increase the quality of the presented manuscript. I highly suggest that the authors rephrase/specify/add the following:
- Please elaborate on why the authors’ performed a literature review instead of a meta-analysis (e.g. because it was impossible due to the different operationalizations of sexting and related variables). The authors write “Fifthly, due to high levels of heterogeneity, meta-analytic review techniques were not appropriate and hence this review presents conceptual findings, but cannot comment on statistical findings.” This sentence is not correct. The meta-analytic approach was appropriate, but it was impossible due to various reasons.
Thanks for this note. The authors agree with the reviewer. Meta-analysis was considered, but was deemed impossible due to the heterogeneous operationalization of sexting and related variables, and the varied study designs. A clarification to that effect has been added to the last lines of section 2.2. Search Strategy.
- Please explain why the authors opted out of including sexting synonyms in their search strategy. Many sexting studies are hidden behind similar or broader terms, such as cybersex, cybersexuality, netsex, online sex, distance sex, etc. I believe that the authors’ approach with the usage of only those keywords presented in Table 1 could largely limit the possible results they could get. What was the methodology behind selecting the used keywords?
Thank you for this statement. We agree that other terms are often used in broader conceptualisation such as titles and content, and these broader synonymous terms of sexting weren’t included during the search. For thoroughness during this review process, the first author conducted a search using these broader synonymous terms in the following databases: Academic Search Complete, CINAHL, PsycINFO, PsycArticles, PsycExtra, and Psychology and Behavioural Sciences Collection. No additional longitudinal studies were identified as a result of the search. The authors submit that it is likely the initial search terms proposed in the review were sufficiently robust, and that use of the term “sexting” and associated synonyms as indicated in Table 1 within title, abstract and keyword fields, along with referencing previous systematic reviews, was an appropriate and efficient methodology in this case. A clarification has been added to section 2.2. Search Strategy.
- The authors write “A random sample of 10% was reviewed by the second author for verification purposes.” If that approach is in line with current recommendations, please refer to them with a proper literature reference.
Thank you for this statement. There are various recommendations regarding the value of single screening, full double screening or partial double screening, with some systems requiring complete double screening at all stages. However, this approach can be resource intensive, as noted by Waffenschmidt et al (2019). In their methodological systematic review, they found that double screening would have resulted in reduction in missed studies, but that where the reviewer was more experienced, missing studies had no or minimal impact on findings. Overall, Waffenschmidt et al (2019) concluded that any losses in screening accuracy were likely to be marginal, with single screening is justifiable where conducted by an experienced reviewer.
Given limited resources available to complete this review, we adopted an interim methodology, with one reviewer screening all studies. However, as the first author was somewhat less experienced, a second, more experienced reviewer screened a random 10% of eligible papers. As this did not result in any additional papers included for further review, this was deemed a reasonable approach. We have added this information to the manuscript.
Waffenschmidt, S., Knelangen, M., Sieben, W., Buhn, S. & Pieper, D. (2019). Single screening versus conventional double screening for study selection in systematic reviews: a methodological systematic review. BMC Med Res Methodol, 19, 132 https://doi.org/10.1186/s12874-019-0782-0
- In 3.4 authors refer to psychological well-being. It is a widely used term in regard to Carol Ryff’s psychological well-being theory. In 3.4 the authors refer to psychological mental health outcomes such as depression or anxiety, which are not a part of a PWB construct. What is more, in 3.4 authors refer to positive affective states, which are a part of a different theory, the subjective well-being model by Diener. I highly recommend that authors rephrase all PWB-and SWB- related terms into mental health-related ones because, in the current form, the presented paragraph is misleading and not in line with the widely used two theoretical models.
Thank you for this suggestion and insight into theory. As recommended, in order to avoid the conflation of distinct concepts, references to “psychological well-being” have been replaced by “mental health”, while references to “positive/negative affective states/affects” have been replaced with “positive/negative emotions”.
- The authors write “Findings from this review offer a number of implications for researchers and professionals developing education and prevention programs”. As a clinical practitioner and researcher myself, I cannot agree with this sentence. The presented literature search does not provide any guidelines for psychoeducational programs, what is more, it makes it even more difficult because the presented results are not in-line with each other. In its current form, the aforementioned authors’ sentence feels like an obligatory sentence which is present in many manuscripts but does not bring anything to the subject. If the authors believe that the presented results can help in psychoeducational program development, please elaborate and give proper examples of how.
The authors acknowledge and agree with the reviewer’s concerns. For better clarity, we have moved the claim surrounding the “development of education and prevention programs” to the final paragraph of section 4.1. Implications, where there is more support of how findings contribute to the development of prevention programs.
- The authors write that “it is, therefore, necessary to shift prevention efforts away from abstinence-based approaches and towards education on engagement in healthy sexting”. On what bases, do the authors believe that abstinence-based approaches are commonly used? Please refer to a proper scientific publication. In past years, abstinence-based approaches are being waived towards a new and more effective control use model. The whole Implications section feels like it was written by a researcher and scientist rather than a practitioner. I highly recommend that authors back up their examples with examples and practical recommendations because, in the current form, it provides directions for future research.
Thank you for this suggestion. The end of paragraph 1 of section 4.1. Implications has been edited based on recommendations. The authors acknowledge that while important progress is being made in moving from abstinence-only education to effective harm-minimization/control models for adolescents, abstinence-based approaches remain prevalent in secondary schools. Therefore, it is important to reiterate the evidence on sexting normalcy and encourage a move away from abstention.
A recent reference to abstinence-based sex education has been added in under section ‘4.1. Implications’ and also referenced below for the reviewer’s convenience. A more cautious recommendation has been provided, including directions for future research as recommended.
York, L., MacKenzie, A., & Purdy, N. (2021). Sexting and institutional discourses of child protection: The views of young people and providers of relationship and sex education. British Educational Research Journal, 47(6), 1717–1734. https://doi.org/10.1002/berj.3751